# Defect Repair Deposit and Insurance Premium for a New Home Warranty in Korea

Junmo Park 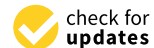 and Deokseok Seo *

School of Architecture, Halla University, Wonju-si 26404, Republic of Korea
* Correspondence: seodk@halla.com; Tel.: +82-10-2289-9946

**Abstract:** Disputes due to defects in newly built houses are increasing worldwide. A house builder is responsible for repairing any defects in a newly built house. However, since house builders' risk of closure and bankruptcy are increasing due to aggravated disputes and economic crises, builders may become insolvent and fail to perform defect repairs. In preparation for this, many countries have established defect repair deposit or guaranty insurance systems; however, the standards for these systems are not based an objective evidence since the current standards were arbitrarily established during industrialization. It has been pointed out that Korea's housing defect repair deposit has been set excessively high and is being abused in disputes. Based on dispute cases in Korea, this study analyzed housing construction costs, deposits, and defect repair costs, resulting from lawsuits due to defects. The results confirmed that the defect repair deposit has been set too high compared to incurred defect repair costs. In addition, it was found that the guaranty insurance premium in lieu of the housing defect repair deposit was excessive compared to the damage caused by builder insolvency. In order to improve this, in this study, we proposed two alternative plans in which the housing defect repair deposit was set at a certain percentage of the construction cost based on the current Korean standard. In addition, based on the concept of different deposit levels using the scale of housing construction, such as in Australia and Canada, two additional alternative plans with different deposit ratios for each scale of housing construction were presented. The comparison results for housing defect repair deposits and guaranty insurance premiums based on the four presented alternative plans accompanied by actual cases showed that all the alternative plan deposits were higher than the actual defect repair costs. Even in the case of a guaranty insurance premium, the level was at least twice as high as the damage caused by builder insolvency. Therefore, all the alternative plans can fulfill their original role of protecting homeowners in the case of builder insolvency. At the same time, reducing the guaranty insurance premium to reflect the cost of housing construction is possible, and would benefit both house builders and home buyers. The results of this study are valuable as a reference for other countries considering establishing or revising a housing defect repair deposit system. Specifically, these findings, which analyzed the case of Korea's socioeconomic changes as it transitioned from a developing country to a developed country, can provide important information for many developing countries operating housing defect repair deposit policies and systems.

**Keywords:** defect repair; new home warranty; deposit; insurance premium

## 1. Introduction

Quality problems such as cracks and leaks in newly built houses are called defects. If the homeowner asks for repairs, the house builder is responsible for repairing them [1]. However, even with the current advanced technologies in the construction industry, many defects are occurring in house construction [2], which have resulted in disputes among developers and between house builders and homeowners [3]. If difficulties in the business, including these disputes, are aggravated, housing builders may become insolvent due to business closures, bankruptcies, dissolutions, etc. [4], and, in these cases, homeowners do

not receive compensation or protection for physical and economic losses caused by defects that occur regardless of their warranty or deposit [5].

Every country prepares against housing problems in the form of a housing defect repair deposit or new home warranty—a guaranty to repair defects due to builder insolvency [6–12]. Regarding the value of the deposit, in some cases, a specific range is proposed [8–10] and, in other cases, there is no ceiling set [11,12]. Alternatively, the value of the deposit may be calculated based on the construction cost [8,11,12] or in proportion to the number of households [9,10].

In Korea, it is stipulated that the housing defect repair deposit is 3% of the housing construction cost [12]. Since the deposit is reserved for defect repairs, the value of the deposit must be negotiated at an appropriate level after finding out the incurred costs. However, the standards for Korea's defect repair deposit have been calculated without specific evidence, and the deposit system lacks a logical rational. Since Korea's standards were prepared at the beginning of industrialization, sufficient base data were unavailable. Similar to most countries that have gone through industrialization, the deposit system in Korea has not been uniformly prepared using political judgment with administrative convenience.

According to prior studies, defect repair costs can be a minimum of 0.18% to a maximum of 10.27% of the construction cost [13–19]. In addition, Park and Seo analyzed cases of Korean housing and found that the average defect repair cost was less than 1% of the construction cost, and even in the highest case, it did not exceed 2.5% [20]. Based on the results of the above case studies, Korea's housing defect repair deposit has been set at an excessive level. Therefore, the standard for the housing defect repair deposit needs to be adjusted to an appropriate level. If the excessively high housing defect repair deposit was appropriately reduced, it is expected that deposits as well as guarantry insurance premiums, which are the fees received by an insurer when issuing insurance in lieu of a deposit, would be reduced. If housing defect repair deposits and guaranty insurance premiums were reduced in such a way, housing construction costs would also be reduced, which would be an economic benefit for home buyers.

However, previous research has not suggested how much the housing defect repair deposit and guaranty insurance premium could be reduced. In addition, even if the housing defect repair deposits and guaranty insurance premiums are reduced, it is necessary to examine whether reducing deposits and premiums would cause problems in the operation of the housing defect repair deposit system.

In order to supplement the results of earlier research, this study provides empirical evidence on how much the deposit can be reduced and whether the existing defect repair cost can be afforded, and, at the same time, introduces an alternative plan for the defect repair deposit. Specifically, this study contributes to knowledge related to defect repair deposit standards by discussing three major research issues. First, if the defect repair deposit standard is revised, how much can be saved by applying the revised deposits and guaranty insurance premiums? Second, can the revised deposit rate cover the repair cost? Third, can the income from the revised guaranty insurance premium cover the damage caused by builder insolvency?

To answer these questions, in this study, the approach is as follows: The defect repair deposit alternative plans suggested in previous studies and one added alternative plan described later in Section 3 are compared with the current standards. The defect repair deposit alternative plans are applied to defect litigation cases, and deposits and guaranty insurance premiums are calculated accordingly. Based on the calculated deposits and guaranty insurance premiums, an empirical analysis is conducted to compare them with repair costs, especially when a house builder is insolvent.

This paper is organized as follows for the above comparison: In Section 2, hypotheses for the research object are discussed through a literature review. In Section 3, we describe the framework and methods for conducting research. In Section 4, we report the data analysis and comparison results for the research questions. Section 5 provides a comprehensive

discussion of the research questions and analysis results. Finally, in Section 6, we describe the limitations of this study, along with conclusions for future research.

## 2. Literature Study

### 2.1. A New Home Warranty

Defects that occur while constructing buildings, including houses, originate from various causes. In some instances, defects may occur due to inappropriate laws and standards, errors in design [21], errors in material selection or procurement [22], and faulty materials or designs during construction [23]. Problems that occur while using a house after it is completed and delivered to a buyer also involve various defects [24]. Therefore, it is difficult to conclude that all quality problems in newly built houses are defects caused by the house builder.

However, it is expected that most of the defects in a newly built house are the responsibility of the house builder. Therefore, housing-related laws principally stipulate that a house builder is responsible for repairing defects in a newly built house. Meanwhile, there is a supplementary system for cases where house builders cannot repair the defect. This system is applied for cases where a house builder becomes insolvent and cannot repair defects. Although there are differences from country to country, house builders are required to deposit security in financial institutions or purchase guaranty insurance [6–12]. Therefore, to prepare a deposit or guaranty insurance premium for repairing housing defects, it is necessary to predict how many defects may occur in the house and how much it will cost to repair them, and then prepare appropriate premium levels accordingly.

According to related studies, based on the average repair cost/construction cost ratio, from the lowest to the highest value, the following analysis results have been presented: 0.18% of the construction cost is required for defect repairs according to Love et al. [13], 0.538% of the construction cost is required for defect repairs according to Park and Seo [20], 1.10% of the construction cost is required for defect repairs according to Choi [14], 2.75% according to Forcada et al. [15], 4% according to Mills et al. [16], 4.40% according to Josephson et al. [17], 4.95% according to Liu et al. [18], and 5.4% according to Hwang et al. [19]. The above levels suggest that, on average, between 0.18% and 5.4% of the house construction cost is required to repair defects.

In addition, Park and Seo presented a rough figure of the appropriate level of deposit through an analysis of records about housing defect repair deposits in Korea [20]. According to this analysis, the defect repair cost, in 93% of all cases, was less than 1% of the construction cost, and therefore, the current standard of 3% of the construction cost was found to be excessive. Accordingly, it was suggested that the standard for the housing defect repair deposit could be revised to 1% of the construction cost. In addition, an alternative plan to reduce the housing defect repair deposit to 1.5% of the construction cost was proposed as a careful approach to improving the deposit system.

The two above alternative plans presented by Park and Seo were standards for collectively applying a rate lower than the current standard, but when they were applied, the repair cost exceeded the deposit in some cases [20]. Therefore, as another alternative plan to supplement this, Park and Seo suggested a rate of housing defect repair deposit according to the scale of the construction cost. This alternative plan consists of a total of three groups based on the scale of construction. By applying the exchange rate between the Korean won and the USD as of the end of June 2022, 2.5% of the construction cost is required as a deposit if the construction cost is less than USD 62 million, 1.5% is required a deposit if the construction cost is from USD 62 to 125 million, and 1% is required as a deposit if the project is more than USD 125 million. This alternative plan seems most reasonable in terms of protecting the homeowner, as there has been no case where the repair cost exceeded the deposit. Australia [8], Canada [9], and Japan [10] are representative countries that apply differential deposits according to the construction cost or the number of households in the housing defect repair guaranty system.

The contributions of the preceding studies highlighted the need to improve the housing defect repair deposit system. However, no empirical evidence was presented to determine the effectiveness of the defect repair deposit system improvement. Although justification for improving the housing defect repair deposit system has been established, an empirical analysis of the alternative system's expected effects needs to be followed up.

*2.2. Importance of a Warranty*

As mentioned above, although it is different in each country, a housing defect repair deposit can be deposited in cash at a financial institution or replaced with the issuance of guaranty insurance. However, it is rare for a house builder to deposit a defect repair deposit in cash at a financial institution, and it is common to provide guaranty insurance instead of a deposit. Since the house builder does not deposit a bond other than the guaranty insurance issued by the guarantor, there is no refund even after the guaranty period ends. In other words, the housing defect repair deposit system has the deposit as a nominal standard. The guaranty insurance company operates the system using the received insurance premium after issuing the guaranty insurance. Therefore, the two following conditions must be met to amend the housing defect repair deposit system. First, the defect repair deposit is a nominal standard and, if it is reduced, it must be able to afford the defect repair cost. Second, the guaranty insurance premium is a practical standard and, if it is reduced, it is necessary to determine whether a house builder can afford the damage caused by the insolvency.

An insurer prepares for a guaranty insurance case with the guaranty insurance premium received in exchange for providing insurance against defects to the house builder. If a situation arises in which a house builder is insolvent, the insurer must pay the homeowner the cost of repairing defects as a benefit; there is no problem if the insurance money paid is within the range of the guaranty insurance premium. However, if the insurance money paid by a guaranty company is greater than the guaranty premium, it may adversely affect the company's financial condition, making it difficult to operate the system. In this case, the insurer may increase the guaranty insurance premium or refuse to issue defect repair insurance, and the insurer may even withdraw from the insurance business or go bankrupt.

Considering these points, one must look at the situation faced by house builders who need to purchase housing defect repair insurance. The COVID-19 pandemic, which began in 2019, is impacting the entire world. Although the situation in each country is slightly different, it is believed that each country's economy has fared better than expected [25]. According to the World Bank statistics, in 2009, during the global economic crisis due to the subprime mortgage crisis of Lehman Brothers, the world GDP growth rate was −1.3%, but it converted to 4.5% in 2010. In addition, in 2020, when the COVID-19 crisis began to affect the economy directly, the global GDP growth rate was −3.3%, but it converted to 5.8% in 2021 [26]. However, from 2021, when the rapid spread of COVID-19 was controlled to some extent, economic chaos was caused by rapid inflation due to the accumulated global quantitative easing [27]. In order to control this, in 2022, each country's central bank interest rate was raised, the economy experienced a recession [28], and the risk of builder bankruptcy began to materialize.

In the case of England, in 2022, many house builders went bankrupt such as Crossfield, South West Construction Firm, and Beaumont Morgan Development [29]. In the case of the United States, Dirtmaster, Paz Construction, Decotal Construction and Remodeling, etc., filed for bankruptcy, and it has been reported that rehabilitation will be difficult due to enormous debts [30]. In the case of Canada, it has been reported that, as of the end of May 2022, the risk of insolvency in the construction industry increased by 33% compared to the previous year [31]. In Japan, it has been reported that the number of bankruptcies in the construction industry increased by 29.8% compared to 2021 [32]. In the case of Korea, concerns over the financial soundness of large construction companies have also been growing, and it has been reported that Lotte E&C, one of the top house builders, is

promoting a large-scale paid-in capital increase [33]. Therefore, the risk of insolvency of house builders is a common problem worldwide.

Next, regarding the situation of guaranty companies that provide home defect repair insurance, in the past, it has been reported that a representative home guaranty company went bankrupt in the United States [34], and, recently, two guaranty companies in Colorado were reported to have gone bankrupt [35]. A case of bankruptcy of a housing guaranty company has been reported in Australia [36], and a similar case has been reported in New Zealand [37]. In Korea, during the Asian financial crisis of the 1990s, 5 out of 33 banks were liquidated, and 5 other banks were merged [38]. In addition, during the global financial crisis in the 2010s, 30 mutual savings banks were suspended due to insolvency [39]. Korea is also currently experiencing an economic crisis due to the coronavirus pandemic that started in 2019 [40]. Fortunately, there has been no report of a guaranty company bankruptcy, especially a financial institution that performs a housing defect repair guaranty business.

Previous studies have reported on the influence of various aspects of the global economic crisis. Among the housing-related indicators, representative indicators include the impact of mortgage interest rates and inflation [41], changes in the ratio of consumption expenditure to income, and oversupply problems [42]. The current crisis, due to the complex impact from the coronavirus and economic problems, is unprecedented, and it is difficult to predict whether it will expand or be limited to some damage. However, it is clear that, as the economic crisis continues, the risk of insolvency increases for housing businesses, and even guaranty companies can go bankrupt.

If the number of insolvent house builders increases, the benefits that insurers have to repay will increase. Particularly, if an insurer's insurance payments exceed the guaranty insurance premium earned, it may have a negative impact on the insurer's management. Therefore, it is important to determine how the housing defect repair guaranty program operates, the value of guaranty insurance premiums earned, and whether sufficient levels are accumulated in preparation for insurance payments.

*2.3. Warranty Programs*

Financial institutions such as guaranty companies and insurance companies set guaranty insurance premiums by taking into consideration whether the risk of paying insurance money to the policyholder is high or low. Concerning the housing defect repair guaranty, various conditions of the house subject to the guaranty and the house builder who intends to purchase guaranty insurance are evaluated to determine the guaranty insurance premium accordingly. There are various standards for calculating guaranty insurance premiums for housing defect repair, and representative programs in each country are described below.

In the case of United States, the Residential Warranty Company is a representative housing guarantee company that provides a 10-year warranty program for new homes, in which the premium is calculated by multiplying the ratio per USD 1000 by the total price that evaluates the number of houses subject to warranty, the average sale price, and the contractor's performance [43]. In the case of Australia, according to the program used called icare, housing construction-related insurance is subject to a discount rate based on type of construction, contract amount, housing location, and house builder [44]. The range of the discount rate calculated using icare is ±30%, and the contract amount is additionally reflected to calculate the guaranty insurance premium [45].

In the case of Ontario, Canada, a guarantee program is offered by Tarion, a provincial government-affiliated home warranty management agency. Tarion's warranty enrolment fee depends on the value of the assessed home, with minimum and maximum enrolment fees of CAD 330 and CAD 1745, respectively. In addition, since 2021, in Ontario, the premium is calculated by adding a regulatory oversight fee according to the Home Construction Regulatory Authority (HCRA), which strengthens consumer protection [46]. The National House Building Council (NHBC), an organization of British housing companies, determines the levels of insurance premiums by evaluating how long the builder has been

a member of the association and how many disputes have been filed against that house builder [47].

There are five housing defect guaranty companies in Japan, and the maximum payment amount is JPY 20 million, but each company has a different insurance premium calculation method [48]. In the case of Jutaku Anshin Warranty Ltd., the premium consists of additional insurance premiums added to a basic premium, including reinsurance premiums, dispute settlement charges, intentional and gross negligence damages, etc.; an investigation fee is added to calculate the final insurance premium [49]. Another insurer, the Organization for Housing Warranty Ltd., calculates the premium by adding a fee to the basic insurance premium based on the number of inspections. Detached houses are inspected twice, apartments are inspected three times, and the unit price of the basic fee and inspection fee is also different [50].

### 2.4. Warranty Companies in Korea

Although the housing defect repair deposit system was enacted in Korea years ago, the appropriateness of the deposit standard has not been verified. In fact, until the 2010s, there had been no effort to improve the overall housing system. As housing defect lawsuits increased rapidly, some systems, such as defect judgment standards and defect liability periods, were improved in the 2010s [51,52]. Nonetheless, improving the housing defect repair deposit system has not been pursued further. However, in recent years, as some studies have been conducted, it has been confirmed that the deposit standard is excessive [14,20]. Meanwhile, there has been no discussion about the guaranty insurance premium attached to the housing defect repair deposit. Therefore, in this section, we discuss Korea's guaranty insurance for housing defect repairs.

In Korea, several guaranty companies provide defect repair guaranty insurance to housing project owners. Among them, the Korea Housing and Urban Guaranty Corporation (KHUGC), the Construction Guarantee Cooperative (CGC), and the Seoul Guaranty insurance Corporation (SGIC) are representative companies that have the largest business scale [53–55]. Guaranty insurance companies in Korea calculate the guaranty insurance premium by multiplying the housing defect repair deposit by the rate through the contractor's corporate credit evaluation and the guarantee period. The rate of guaranty insurance premium is around 1% of the deposit.

Since the house builder is obligated to make the defect repair deposit, the house builder pays the guaranty insurance premium, and there is no case where the home buyer or owner separately pays the premium. In addition, the guaranty insurance premium is set for each detailed construction, such as reinforced concrete, finishing, facility, and landscaping. The guaranty insurance period varies from 1 to 10 years, but the guaranty insurance premium is paid in a lump sum at the time of subscription. Korea's housing defect repair deposit is set at 3% of the construction cost and can be calculated using Equation (1), and the guaranty insurance premium is calculated using Equation (2) by multiplying the deposit, the guaranty insurance premium rate, and the insurance period as follows:

$$\text{Deposit} = \text{Construction cost} \times 0.03, \tag{1}$$

$$\text{Insurance premium} = \text{deposit} \times \text{insurance premium rate (\%)} \times \text{insurance period (year).} \tag{2}$$

The housing defect repair and warranty programs in each country reviewed above are summarized in Table 1 below. These countries have different standards for home warranty coverage. Notwithstanding, it is to be noted that the current price or current condition of the house subject to guarantee is investigated and reflected in the calculation of the guaranty insurance premium, and the performance or reputation of the house builder is considered. However, Korean insurance premium standards that do not reflect these points could differ from those of other countries.

**Table 1.** Comparison of warranty programs.

| Nation | Estimation Factor To Insurance Premium |
| --- | --- |
| United State of America | Number of homes, average sale price, a record of the contractor |
| Australia | Construction type, contract value, location of the home, icare builder rating |
| Canada | Home sales price, enrolment fee, HCRA regulatory oversight fee |
| United Kingdom | Duration of enrolment to the association, number of claims |
| Japan | Enrolment fee, inspection fee, reinsurance premium, dispute resolution fee, amount of intentional gross negligence damage |
| South Korea | Deposit, warranty fee rate, duration of the warranty |

### 2.5. Role of the Korean Warranty Companies

Construction insurance is a strategy that promotes development of the construction industry, and it has been recommended that it could be applied at the global level [56]. In the case of Korea, most houses are built as pre-sale houses, in which contracts are established and some of the construction costs are paid before construction begins [57]. Since a pre-sale is a contract for a house without a physical house, it is accompanied by a sale guaranty as a protective measure for the home subscriber [58]. Such a housing-related guaranty is a social safety device to protect weaker home subscribers. From the same point of view, a housing defect repair guaranty can be considered to play a public role in protecting socially underprivileged homeowners. The purpose of public institutions is to protect, maintain, and promote shared values [59].

From this point of view, the characteristics of Korean guaranty companies are summarized in Table 2. The Ministry of Land, Infrastructure, and Transport, a Korean government agency, is a major shareholder with a 70.25% stake in the Korea Housing and Urban Guarantee Corporation. In addition, it provides guarantee programs for the construction of houses and cities, as well as sales under the Housing and Urban Fund Act [60]. The Construction Guarantee Cooperative is a cooperative that is contributed to and operated by house builders. Although the specific governance structure is not disclosed, they are supervised by the Ministry of Land, Infrastructure, and Transport following the Framework Act on the Construction Industry [61]. The Seoul Guarantee Insurance Corporation was revived with support from public funds under the Special Act on the Management of Public Funds, due to the 1997 foreign exchange crisis in Korea. According to the summary of chief management status for 2021 disclosed by the Seoul Guaranty Insurance Corporation, the major shareholder is the Korea Deposit Insurance Corporation, a public corporation under the Ministry of Economy and Finance, a Korean government agency, with 93.85% of the total stake [62].

**Table 2.** Comparison of the Korean warranty companies.

| Warranty Company | Basis of Establishment | Type | Major Shareholder | Shareholding Rate of Major Shareholders |
| --- | --- | --- | --- | --- |
| KHUGC | Housing and Urban Fund Act | Public corporation | Korean government | 70.25% |
| CGC | Framework Act on the Construction Industry | Mutual aid association | Undisclosed | Undisclosed |
| SGIC | Special Act on the Management of Public Funds | Public corporation | Korea Deposit Insurance Corporation | 93.85% |

Taken together, Korean housing guarantee companies are public institutions that perform a public duty of guaranteeing housing defects. Therefore, these housing guaranty companies should focus on realizing the public's social interest rather than pursuing profit in operating the housing defect repair guaranty system. Therefore, it is reasonable that Korea's housing defect repair insurance premium should be set at an appropriate level to realize the public interest.

*2.6. Metrics and Evaluation of Insurance Premiums*

Regarding the guaranty insurance premium that an insurer receives while providing a housing defect repair guaranty and whether the premium level is appropriate, management-related data such as financial statements and audit reports published by three Korean guaranty companies did not disclose details of housing defect repair guarantees [53–55]. Therefore, since it is currently challenging to grasp the overall premium levels of housing defect repair guaranty insurance, it is practical to select a method of inferring the guaranty insurance premiums paid for by the cases.

In addition, the standards and criteria have been examined to see if the guaranty insurance premium for the housing defect repair deposit is appropriate. It is common in the insurance industry to use the insurance loss rate as a standard rate between the insurance premium and the insurance payment [63,64]. Therefore, the insurance loss rate can also be applied to the insurance assessment of the housing defect repair guarantee. However, as mentioned above, three Korean guarantee companies do not disclose insurance loss rates for housing defect repair guarantees [53–55]. Therefore, there is a need to identify a method of indirectly inferring the level of the insurance loss rate of the housing defect repair guarantee through the cases.

Since there is no information about the insurance loss rate of the housing defect repair guaranty insurance, other types of insurance were looked at. In the case of automobile insurance, according to the literature, as of 2018, the insurance loss rate was 58.5% in Japan, 61.7% in California, America, 69.5% in the United Kingdom, 82.4% in Korea, and 85.5% in Germany [65]. In addition, considering the situation in Korea alone, between 2016 and 2018, the insurance loss rate, on average, was 81.8% for long-term insurance, 75.9% for automobile insurance, and 58.7% for non-life insurance [66]. Therefore, all the types of insurance have various insurance loss rates depending on the country and subject. Although it is difficult to directly compare the insurance loss rate for housing defect repair guaranty insurance with other types of insurance, since there is no adequate standard for judging the insurance loss rate of the housing defect repair guarantry level, the comparison results are utilized as references.

In this study, based on practical cases under the system of Korea, the balance between the premium income for housing defect repair guaranty and the insurance payment confirms any problem with the soundness of guaranty insurance. In addition, using the insurance loss rate as an indicator is intended to determine whether the risk of loss in housing defect repair insurance is higher than that of other types of insurance. Through this comparison, we can evaluate whether the alternative plans for improving the housing defect repair deposit standards are appropriate. Furthermore, this comparison study will provide empirical evidence of the cost savings of applying each alternative plan to a housing defect repair deposit.

## 3. Materials and Methods
### 3.1. Framework

This study was conducted according to the framework shown in Figure 1. The specific targets, as well as data collection, analysis, and comparison according to the framework are sequentially described in the following sections.

The framework for this study is as follows:

(1) Data collection: The defect repair cost, security deposit, construction cost, and repair cost data, in the case where there is builder insolvency, are collected from cases of housing defect lawsuits.

(2) Selection of deposit alternatives: Three alternative plans proposed in previous studies and one additional alternative plan proposed in this study are adopted. The current standards and the four alternative plans are compared.

(3) Calculation of deposit and guaranty insurance premium: The deposit and guaranty insurance premium based on the data in (1) about the current standard and alternative plans in (2) are recalculated, and this is used to calculate the defect damage rate,

insolvency damage rate, and insurance loss rate. The guaranty insurance premium is calculated by applying the insurance premium rate of the Korea Housing and Urban Guarantee Corporation, a representative guaranty company in Korea.

(4) Comparison of deposits of alternatives: The deposits and defect repair costs of the four alternative plans recalculated in (3) are compared. If the deposits of all the alternative plans are less than the defect repair cost, these alternative plans are inappropriate as a standard, and the deposits and other miscellaneous items must be recalculated after supplementation.

(5) Comparison of guaranty insurance premiums of alternatives: The guaranty insurance premiums of the four alternative plans recalculated in (3) and the repair cost in the case of insolvency are compared. If the guaranty insurance premiums of all alternative plans are less than the repair cost of builder insolvency, these alternative plans cannot protect the insolvency case. Therefore, the guaranty insurance premium should be recalculated.

(6) Proposal of final alternative: It is proposed to select a reasonable and stable alternative plan by comparing whether the alternative plans for deposits and guaranty insurance premiums are appropriate.

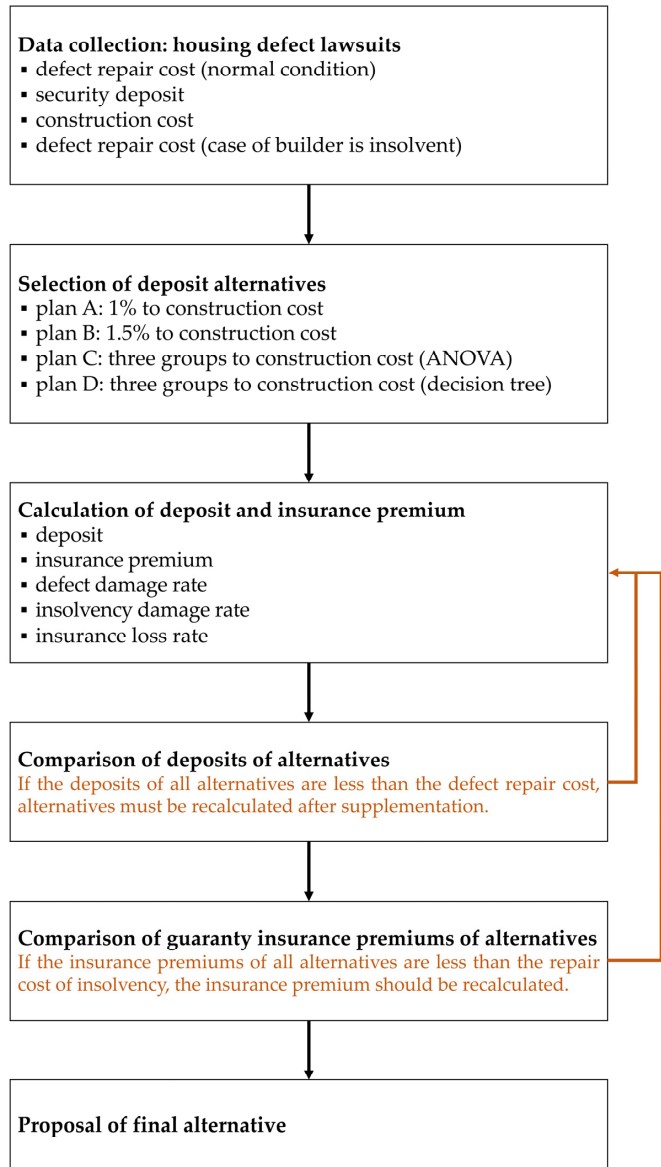

**Figure 1.** Research framework.

### 3.2. Object and Scope

Housing, the subject of this study, means a newly built apartment house in Korea. Construction cost (CC) is a directly used cost and does not include land or financial costs. In most cases, the housing construction cost is considered to be equal to the contracted construction cost agreed to by the developer, the builder, and the general contractor (GC).

Defect repair cost (DRC) refers to the cost recognized by the court as necessary for repairing defects through a lawsuit. The court selects a professional engineer or certified architect as an appraiser; the appraiser visits and inspects the house that is the subject of the lawsuit, to calculate the cost required for repair.

The housing defect repair deposit is set at 3% of the construction cost, following the Korean Housing Act. The Korean Housing Act classifies 18 types of detailed work, including reinforced concrete, finishing, landscaping, window, waterproofing, water supply, and electrical works, which are put into housing construction. In addition, the defect liability period for each specialized work is divided into 1 year, 2 years, 3 years, 4 years, 5 years, and 10 years. This defect liability period is applied mutatis mutand is as the warranty period for housing defect repair guarantees [67].

The housing defect repair insurance premium (IP) refers to a fee paid to a guaranty insurance company by a developer or constructor, who is a builder, when issuing insurance from a guaranty insurance company instead of depositing cash. Insurance payment refers to the defect repair costs paid by an insurer to a householder when a builder becomes insolvent and is unable to perform the defect repair.

The insolvency of a builder means that it is in a state of closure, bankruptcy, etc., and cannot perform repairs for defects. The insolvent state of a builder is judged by reviewing the asset status, debts, and tax payments in a lawsuit. Therefore, in this study, cases in which the builder is insolvent refers only to cases where the court recognizes that the builder is insolvent at the conclusion of the lawsuit, and it is judged by the insurance payment paid by the guaranty company for the repair in the case of builder insolvency.

### 3.3. Data Collection

This study was conducted on 290 cases in which lawsuits were filed to repair house defects. The court's litigation results are notified to the litigation parties and their legal representatives in the form of a judgment, and other litigation information, can be found on the court's website in Korea [68]. This judgment stipulates the housing defect repair deposit, defect repair cost, and the builder insolvency status.

As mentioned above, since the deposit is set at 3% of the construction cost, the construction cost can be calculated as in Equation (3) using the deposit information in the judgment. In addition, the DRC rate to the CC can also be obtained using the following Equation (4):

$$\text{Construction cost} = \text{Deposit} \div 0.03, \tag{3}$$

$$\text{DRC rate to CC} = \text{Defect repair cost} \div \text{Construction cost} \times 100 \ (\%) \tag{4}$$

### 3.4. Alternative Plans for Deposit

Regarding the housing defect repair deposit standard, this study compares four alternative plans with the current standard by adding three alternative plans presented in a previous study by Park and Seo [20] and the following alternative plan. To distinguish each alternative plan, the proposed deposit of 1% of the construction cost was named Plan A, the proposed deposit of 1.5% was named Plan B, and the alternative of varying the ratio according to the size of the construction cost was named Plan C. Ultimately, the alternative plan added in this study was named Plan D.

The reason Plan D was added to this study is as follows: According to the initial study [20], the defect repair cost ratio of the recorded data was, on average, 0.538% of the construction cost, which was less than 1%; therefore, based on this, Plan A was presented. However, when examined individually, some cases exceeded 1% of the CC. A compromise proposal of 1.5% of the construction cost was also suggested in the previous study to

supplement this point. In addition, since Plan C of the previous study varied the ratio according to the size of the construction cost, the repair cost must not exceed the deposit. However, since it was necessary to present options to be reviewed in the policy decision process for defect repair deposit, Plan D was added as a more prudent compromise plan for Plan C.

In the case of Plan C, there was a difference in the repair cost ratio between the three groups through ANOVA, and Plan D was divided into three groups through a decision tree. As shown in Figure 2 below, according to the analysis results using the SPSS program, Plan D was classified into three groups according to the repair cost ratio. It was the same as Plan C, except there was a difference in the size of the construction cost, which was the criterion for classifying each group. Plan C was classified into less than USD 62 million, USD 62–125 million, and more than USD 125 million. Plan D was divided into less than USD 74.8 million, USD 74.8–159.7 million, and more than USD 159.7 million. In the statistics shown in Table 3 below, the minimum and maximum values for each group do not change, but the average is slightly lower. There is also a change in the number of cases belonging to each group. Due to this difference, the deposit and guaranty insurance premiums are expected to change. In this case, we examined if the deposit exceeded the reasonable repair cost and if the builder could afford the repair cost when it becomes insolvent.

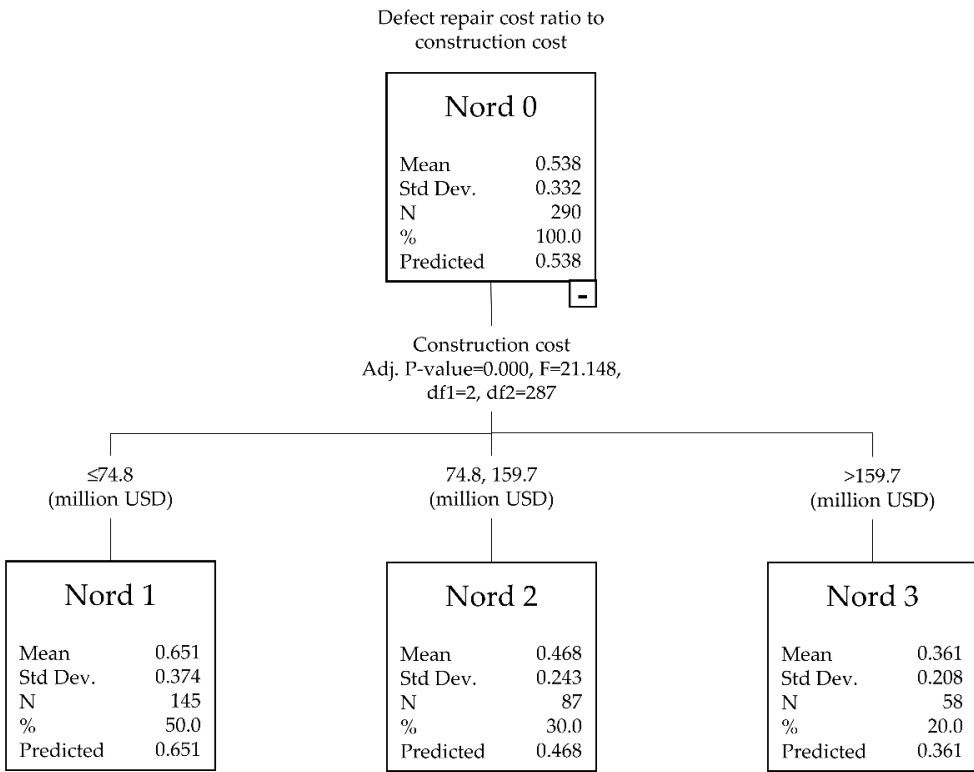

**Figure 2.** Decision tree of defect repair cost ratio to construction cost.

**Table 3.** Statistics of Plan C and Plan D.

| Index | | Plan C: Park and Seo | | | Plan D: This Study | | |
|---|---|---|---|---|---|---|---|
| | | Less than USD 62M | USD 62–125M | More than USD 125M | Less than USD 74.8M | USD 74.8–159.7M | More than USD 159.7M |
| DRC ratio to CC | Max | 2.22% | 1.47% | 0.93% | 2.22% | 1.47% | 0.93% |
| | Mean | 0.67% | 0.50% | 0.39% | 0.65% | 0.47% | 0.37% |
| | Minimum | 0.09% | 0.00% | 0.03% | 0.09% | 0.00% | 0.03% |
| No. of cases | | 117 | 92 | 81 | 145 | 87 | 58 |
| Suggested DRC ratio | | 2.5% | 1.5% | 1.0% | 2.5% | 1.5% | 1.0% |

*3.5. Comparison*

In this study, these values were recalculated based on the recorded data of litigation cases to compare the current deposit standard and the deposits of each alternative plan. The current standard was calculated by applying a ratio of 3% to the construction cost, 1% to Plan A, and 1.5% to Plan B. Plan C and Plan D were calculated by applying 2.5%, 1.5%, and 1% of the construction cost ratio for each construction cost scale indicated in Table 3. The recalculated deposit, defect repair cost, and defect repair cost in the case of insolvency were mutually compared.

To compare the level of the deposit recalculated for each alternative plan with the current standard, the percentage of the defect repair cost to the deposit was used, which was named the defect damage rate and was calculated as in Equation (5). To examine whether the deposit for the alternative plan could protect a homeowner from builder insolvency, the percentage of the insured sum of the insolvency case to the deposit was used, called the insolvency damage rate, and it was calculated using Equation (6):

$$\text{Defect damage rate} = \text{Defect repair cost to total case} \div \text{Deposit} \times 100\ (\%) \qquad (5)$$

$$\text{Insolvency damage rate} = \text{Defect repair cost to insolvency case} \div \text{Deposit} \times 100\ (\%) \quad (6)$$

Meanwhile, the guaranty insurance premium received by the insurer when issuing the guaranty insurance instead of the defect repair deposit was calculated as shown in Equation (2) by multiplying the deposit by the guaranty insurance premium rate and the guarantee period. Guaranty insurance premium rates are different for each of the three insurers: the Korea Housing and Urban Guarantee Corporation ranges from 0.142 to 0.997% [51], the Construction Guarantee Cooperative ranges from 0.35 to 0.95% [52], and the Seoul Guaranty Insurance Corporation ranges from 0.153 to 1.019% [53]. The guaranty insurance premium rate is determined through a credit evaluation of the builder who intends to subscribe to the guaranty insurance, but the rates for each insurance company are not disclosed. Therefore, in this study, the guaranty insurance premium rates of the Korea Housing and Urban Guarantee Corporation, which handles the most guarantees, were compared as a standard, and each guaranty premium was calculated by assuming three rates: the minimum rate, the average rate, and the maximum rate. Based on this, the insurance loss rate, which is a comparison measure for the current standard and each alternative plan, is calculated as a percentage of the defect repair cost in the case of insolvency compared to the guaranty insurance premium, and it can be calculated as in Equation (7):

$$\text{Insurance loss rate} = \text{Insurance payment} \div \text{Insurance premium} \times 100\ (\%) \qquad (7)$$

## 4. Results

*4.1. Deposit and Damage*

The analysis results of 290 cases of housing defect lawsuits in Korea showed that the construction cost was a minimum of USD 7.74 million and a maximum of USD 825 million. In addition, as indicated in Table 4, the total amount of deposits under the current standard was USD 981 million. The total defect repair cost was USD 144 million. Four cases among 290 builders and developers who were the project owners were found insolvent, and the repair cost in this case was USD 0.92 million. These deposits, defect repair costs, guaranty insurance premiums, etc., were converted by applying the exchange rate between the Korean won and the USD at 1277.35 Korean won per USD as of the end of June 2022.

Table 4 shows the recalculated deposit for each alternative plan: USD 326 million for Plan A, USD 490 million for Plan B, USD 433.8 million for Plan C, and USD 469 million for Plan D. If listed in descending order, the amount is in the order of Plan A, Plan C, Plan D, and Plan B.

**Table 4.** Statistics of deposit and DRC.

| Metrics | Current Regulation (USD Million) | Plan A (USD Million) | Plan B (USD Million) | Plan C (USD Million) | Plan D (USD Million) |
|---|---|---|---|---|---|
| Deposit of total case | 981 | 326 | 490 | 433.8 | 469 |
| DRC of total case | 144 | 144 | 144 | 144 | 144 |
| DRC of insolvency case | 0.92 | 0.92 | 0.92 | 0.92 | 0.92 |

If the defect damage rate is expressed as a percentage between the deposit and the repair cost, as shown in Figure 3 below, the current standard is 14.68%, whereas it is 44.17% for Plan A, 29.39% for Plan B, 33.2% for Plan C, and 30.70% for Plan D. In other words, even when Plan A, the highest level of deposit, was applied, the defect repair cost was less than half of the deposit. In other words, a deposit that is more than twice the amount of damage is already secured. Therefore, since the deposit is greater than the defect repair cost in all the alternative plans, there will not be any problem in protecting the defect repair damage.

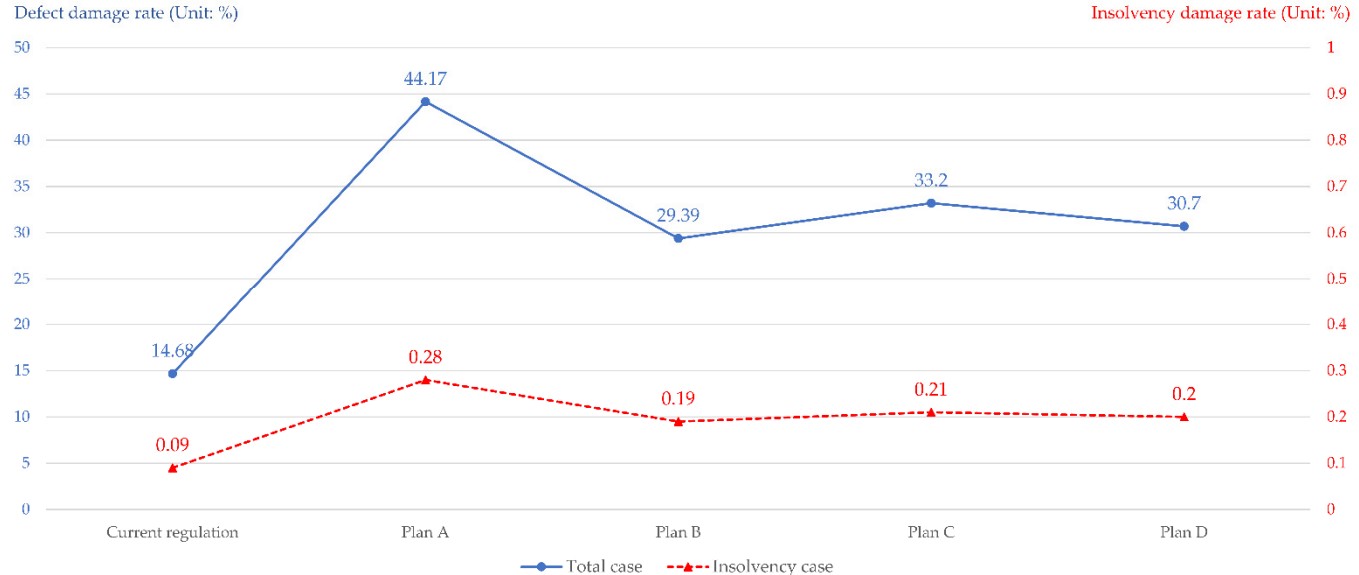

**Figure 3.** Comparison of defect damage and insolvency damage.

In addition, if one looks at the insolvency damage rate expressed as a percentage of defect repair costs in cases where the builder is insolvent compared to the deposit, as shown in Figure 3 below, the current standard is 0.09%, whereas it is 0.28% for Plan A, 0.19 % for Plan B, 0.21% for Plan C, and 0.20% for Plan D. This means that the repair cost due to builder insolvency was less than 1% of the deposit. Therefore, the deposit of all the alternative plans is at a level sufficient to protect the damage caused by builder insolvency.

In this way, for all the alternative plans, the deposit is at a level sufficiently greater than the total repair cost or the builder's insolvency. Therefore, there could be no problem in adopting any of the four alternative plans. Suppose the current standard deposit is set at 100% and compared to each alternative plan. In this case, the repair cost level is only 33.2% for Plan A, 50.0% for Plan B, 44.2% for Plan C, and 47.8% for Plan D. Therefore, the deposit according to the current standard is set excessively compared to the actual defect repair damage, and even if it is reduced by a minimum of 50% to a maximum of 66.8%, the entire repair cost can be sufficiently protected, including the case where the builder is insolvent.

### 4.2. Insurance Premium and Loss

Regarding the insurance premium explained in Section 3, the guaranty insurance premium was calculated by multiplying the deposit by the rate and the guarantee period, which is 1 to 10 years, as set by the Housing Act. The guaranty insurance premium rate was applied according to the credit rating of each insurer based on its standards. Although the minimum and maximum rates were disclosed, each builder's credit rating and guaranty insurance premium rate were undisclosed. Therefore, in this section, the changing trends of the guaranty insurance premiums of 290 cases were compared by considering the minimum and maximum rates disclosed by guaranty companies and the average rate obtained by the arithmetic average of the two.

As shown in Table 5, the total guaranty insurance premium, according to the current standard for defect repair deposits, amounts to a minimum of USD 5.64 million, an average of USD 22.63 million, and a maximum of USD 39.61 million. When each alternative plan is applied, the guaranty insurance premium for Plan A is the smallest compared to the current standard, ranging from a minimum of USD 1.88 million to a maximum of USD 13.2 million. It is followed by Plan C and Plan D, with Plan B's guaranty insurance premiums ranging from a minimum of USD 2.82 million to a maximum of USD 19.81 million.

**Table 5.** Statistics of insurance premium.

| Insurance Premium Ratio | Current Regulation (USD Million) | Plan A (USD Million) | Plan B (USD Million) | Plan C (USD Million) | Plan D (USD Million) |
|---|---|---|---|---|---|
| Max | 39.61 | 13.2 | 19.81 | 17.52 | 18.94 |
| Mean | 22.63 | 7.54 | 11.31 | 10.00 | 10.82 |
| Min | 5.64 | 1.88 | 2.82 | 2.49 | 2.70 |

Therefore, compared to the guaranty insurance premium under the current standard of at least USD 5.64 million, Plan A, which has the lowest guaranty premium among the alternative plans, has a guaranty premium of USD 1.88 million, which is 33.3% of the current standard. Even compared to the guaranty insurance premium of USD 2.82 million of Plan B, which has the largest guaranty insurance premium income, it is only 50% of that amount.

Let us look at the insurance loss rate expressed as a percentage between the guaranty insurance premium and the insurance amount in the case of insolvency of a builder, as shown in Figure 4. The blue line indicates the insurance loss rate when the minimum rate is applied, the green line indicates the loss with the average rate, and the red line indicates the insurance loss rate when the maximum rate is applied. The insurance loss rate is calculated by using Equation (7). On the one hand, the insurance payment (DRC of insolvency case), the numerator of Equation (7), is constant at USD 0.92 million, as shown in Table 4. On the other hand, the insurance premium corresponding to the denominator of Equation (7) varies according to the insurance premium ratio, as indicated in Table 5. For example, in the case of Plan A, the insurance payment is USD 0.92 million, and the insurance premium applying the insurance premium ratio at the highest rate is USD 13.2 million. When substituted into Equation (7), the ratio is 6.97%, as shown in Figure 4.

According to the current standard, the insurance loss rate is a minimum of 2.32% and a maximum of 16.31%. On the one hand, Plan B has the lowest insurance loss rate, ranging from 4.64% to 32.62%. On the other hand, Plan A has the highest insurance loss rate, ranging from 6.9% to 48.94%. No matter which of the four alternative plans is selected, the guaranty insurance premium income of the guaranty company is more than the insurance money in the case of insolvency. Among the four alternative plans, the highest insurance loss rate is 48.94%, calculated by applying the highest guaranty insurance premium rate to Plan A, not exceeding 50%. Therefore, the guaranty insurance premium, recalculated according to each alternative plan, is at least twice as much, in preparation for damage caused by the risk of builder insolvency.

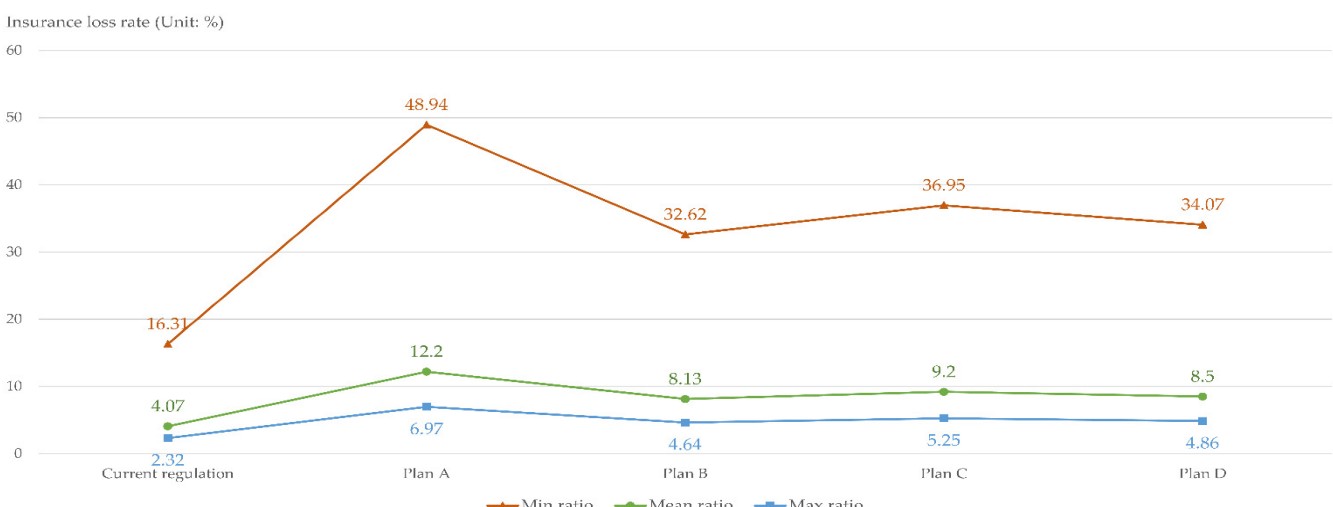

**Figure 4.** Comparison of the insurance loss rates.

The facts above suggest that the guaranty insurance premiums, according to the four alternative plans to the housing defect repair deposit calculation standard, are only from 33.3% to 50% of the current standard. Furthermore, the guaranty insurance premiums of these alternative plans are more than twice the damage due to builder insolvency, and it is considered that there is a sufficient level of margin to cover the actual risks.

## 5. Discussion

### 5.1. Justification to Revise the New Home Warranty

Logical standards based on sufficient evidence must be established to support a system, and any low points must be supplemented. In this regard, Korea's housing defect repair deposit system was established without sufficient evidence at the time of early industrialization; however, over the years, based on the results of prior research and this study, logical alternative plans have been established. The analysis of the deposit and guaranty insurance premium based on the housing defect repair cost performance data showed that the current standard of deposit and guaranty insurance premium is set excessively compared to the required level.

Korea's excessively high housing defect repair deposit might be due to a lack of evidence based on cases to support decision making because it was activated before the housing industry's development. The Korean housing industry took on an industrial remodeling from the 1980s and had a producer-centered industrial structure until the 1990s. From the 2000s, most industries began to be consumer-oriented, and many issues were raised about the quality and defects of housing. In the 2010s, institutional improvements were promoted by the housing industry such as housing defect judgment criteria and a defect liability period. Therefore, it would be an appropriate time for social discussions on housing defect repair deposits.

Housing repair deposits from the perspective of individual projects (project level) and the overall program (program level) have been examined. A review of previous studies at the individual project level showed that the housing defect repair cost averaged 0.538%, less than 1% of the construction cost, and did not exceed 2.5% even in the highest case [20]. In addition, from the perspective of the whole program, as in this study, the total amount of deposit in all cases was USD 981 million, which is 6.81 times higher than the total defect repair cost of USD 144 million. Therefore, the current standards for housing defect repair deposits are sufficient, whether viewed from the perspective of individual projects or the overall program.

Regarding the guaranty insurance premium, which is the basis for practically operating the housing defect repair guaranty system, in Korea, guaranty companies that provide home warranty programs are, in fact, public institutions. These public institutions should

operate with a focus on public interest activities. From this point of view, it is reasonable for the housing defect repair guaranty insurance premium to be set at a level that can prepare for the case where a builder becomes insolvent and fails to perform defect repair. Therefore, we examine whether guaranty insurance premium income is appropriate from the following two points of view.

First, the guaranty insurance premium income received by the insurance company, while providing housing defect repair guaranty insurance to the builder, seems to be sufficient to cover the expenses for the repair of defects caused by builder insolvency. The total amount of insurance paid by the guaranty company due to builder insolvency is USD 0.92 million, whereas the total income is expected to be USD 2.32 million even if the guaranty insurance premium is the lowest under the current standard. Therefore, it is difficult to say that guaranty insurance premiums are insufficient given that the income from guaranty insurance premiums is 2.52 times the amount paid by the guaranty company.

Second, the insurance loss rate of housing defect repair guaranty insurance is not excessive compared to other types of ordinary insurance. Figure 5 shows the insurance loss rate for each type of insurance. As shown in Figure 4, the insurance loss rates of the housing defect repair guaranty insurance are a minimum of 2.32%, an average of 4.07%, and a maximum of 16.31% when the guaranty insurance premium is calculated according to the current standard. Among them, the insurance loss rates were compared, with the highest loss rate at 16.31%. In addition, the loss rates of other types of insurance in the initial study were used for comparison [64]. As shown in Figure 5, the insurance loss rates for different types of insurance are 81.8% for life insurance, 75.9% for automobile insurance, and 58.7% for non-life insurance. On the one hand, the insurance loss rate is high for life insurance and low for non-life insurance.

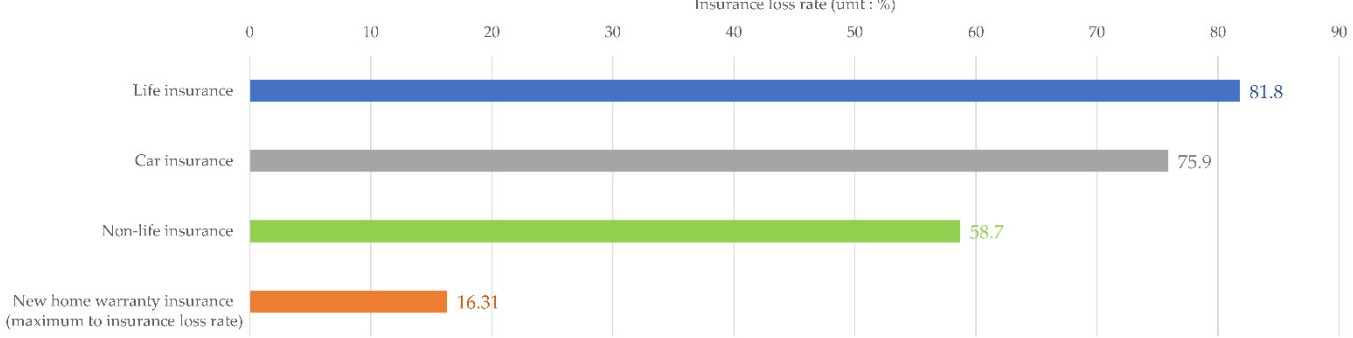

**Figure 5.** Comparison of insurance loss rates among different types of insurance.

On the other hand, the insurance loss rate for housing defect repair guaranty is 16.31%, which is low compared to 28% for non-life insurance. If one considers the insurance loss rate inversely from the perspective of the profit rate through insurance operation, the value obtained by subtracting the insurance loss rate from the total is the profit rate of the insurer. Only 24.1% would be the profit rate for automobile insurance and 41.3% for non-life insurance. It can also be interpreted that the profit rate of housing defect repair guaranty insurance operated by the public is as high as 83.69%.

In summary, according to the current standards, the housing defect repair guaranty insurance premium is excessive compared to the level required in preparation for builder insolvency. Furthermore, in terms of insurance loss rate, the housing defect repair guaranty insurance has a significantly lower insurance loss rate than other types of insurances. Even if the rate is lowered by adjusting the current deposit standard, there will be no problem in the operation of the housing defect repair guaranty system of the guaranty company. Suppose housing defect repair deposits and guaranty insurance premiums are reduced to an appropriate level through this, in this case, it is expected to contribute to the overall economy by reducing construction costs, benefiting home buyers, and reducing unnecessary social costs.

### 5.2. Best Alternative Plan

Since this study aims to discuss justifying improving the housing defect repair deposit calculation standard, it is time to review which alternative plan is reasonable. Four alternative plans were suggested for improving the housing defect repair deposit calculation standard in previous studies and in this study. The exact deposit rate was applied to Plan A, which lowers the deposit from 3% to 1% of the construction cost, and to Plan B, which lowers the deposit to 1.5%. In addition, there is Plan C and Plan D, which have different deposit rates depending on the size of the construction cost. As a result of recalculating the deposit and guaranty insurance premium for each alternative plan, Plan B is the highest, followed by Plan D, Plan C, and Plan A.

Before comparing the alternative plans, we examined whether there were any problems with the deposit and guaranty insurance premium when each alternative is applied. If, in the case of a specific alternative plan, there is a problem with the deposit or guaranty insurance premium, it is difficult to say that the alternative plan is suitable and should be excluded. However, according to the analysis results of this study, there is no problem in terms of the deposit and guaranty insurance premium, no matter which of these four alternative plans is applied. From the viewpoint of the deposit, the sum of deposits of all cases when Plan B was applied, among the four alternative plans, was the smallest, and the amount was USD 326 million.

Since the total defect repair cost for all cases is USD 144 million, the deposit for Plan B is 2.26 times the defect repair cost. Therefore, in terms of the deposit, it is difficult to say that the operating system is complicated with the four alternative plans because the deposit is sufficiently greater than the defect repair cost. Next, as far as the guaranty insurance premium is concerned, the total guaranty premium income when the minimum rate for Plan B is applied is USD 1.88 million. In addition, Plan B's guaranty insurance premiums are greater by more than twice the insurance payments since the total amount of insurance payments due to builder insolvency is USD 0.92 million. Even in the case of applying Plan B, which has the lowest deposit and guaranty insurance premium among the four alternative plans, the deposit and guaranty insurance premium is at least twice as much secure than necessary. It can be concluded that any of the suggested alternative plans do not interfere with the operation of the housing defect repair guaranty system.

If there is no problem with the system operation, even if any of the proposed alternative plans are applied, selecting the alternative plan with the highest system improvement effect would be reasonable. Therefore, how much each alternative plan can save compared to the current standard was investigated. Figure 6 shows each alternative plan's guaranty insurance premium and reduction ratio compared to the current standard for the case where the lowest rate among the guaranty insurance premiums indicated in Table 5 was calculated. If Plan A is applied, the guaranty insurance premium is 33.33% of the current standard, saving 66.67% of the cost. Plan B is expected to reduce costs by 50%, Plan C by 55.85%, and Plan D by 52.13%. Therefore, considering cost reduction, it is reasonable to select Plan A as an alternative to the calculation standard for the housing defect repair deposit.

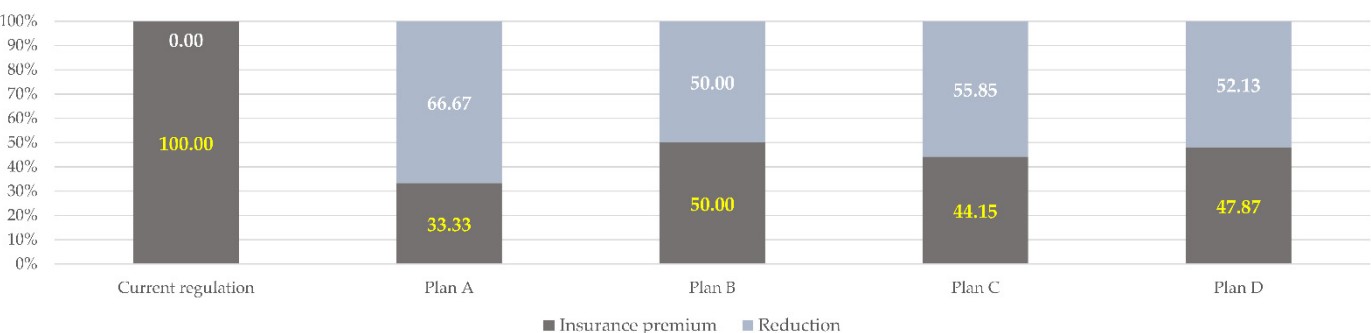

**Figure 6.** Comparison of reduction effect for insurance premium.

Plan A, which reduces the housing defect repair deposit to one-third of the current standard, can be the most innovative in terms of rationality. However, it is also necessary to consider the socioeconomic atmosphere in promoting institutional improvement. From the point of view of the guarantee company, it will likely take a negative stance on improving the system because the earlier stable and low-risk profit will decrease significantly. Moreover, considering that the global economic crisis is escalating from the second half of 2022, it is difficult to ascertain that the insolvency risk of the builders will not increase. It is, therefore, necessary to consider rationality and stability in promoting improvement of the housing defect repair guarantry system. To this end, this study first proposes to create an atmosphere for institutional improvement by adopting Plan B or Plan D, which can be called relatively stable alternative plans, and gradually improve toward Plan A as a mid-to-long-term goal.

## 6. Conclusions

Disputes over the quality and repair of defects in newly built houses are ongoing worldwide. Particularly in Korea, disputes have expanded to litigation, becoming a social problem. The defect repair deposit is one of the most significant issues in these lawsuits. The defect repair deposit or guaranty insurance is a system operated in many countries as a preparation for the insolvency of builders. In the case of Korea, the defect repair deposit is set at 3% of the construction cost. However, it has been controversial because the logic and basis for establishing it have not been presented. In addition, it is generally replaced by guaranty insurance rather than making a deposit, but there has been no verification on whether the guaranty insurance is sufficient to cover builder insolvency. Previous studies have reported cases of various countries on the required scale for repairing housing defects. However, only a few studies have reviewed whether the deposit for defect repair is appropriate, and there has been no research on guaranty insurance.

In this study, the levels of deposit and guaranty insurance in preparation for repairing housing defects were analyzed based on cases in Korea. It was confirmed that the deposit set according to the current standard is excessive compared to the actual defect repair cost. The guaranty insurance premium not only seems reasonable in preparing for damages caused by builder insolvency but also was evaluated as having a low insurance loss rate compared to other insurances. Considering these points, it is reasonable to reduce the housing defect repair deposit and guaranty insurance premium in Korea to an appropriate level since they are set at an excessive level.

Meanwhile, this study suggested four alternative plans to the housing defect repair deposit. No matter which of these alternative plans is adopted, there seems to be no problem in preparing for damages caused by builder insolvency with deposits and guaranty insurance premiums. If the alternative plan proposed in this study is applied, the costs of housing defect repair deposit and guaranty insurance premium are expected to save at least 50%. This reduced cost is reflected in the cost of housing construction, and as housing can be supplied at a lower cost, both the builder and the home buyer benefit. In addition, the action is expected to reduce the wastage of social costs by reducing unnecessary guaranty insurance premiums.

However, this study also has the following limitations. Korea's housing defect repair deposit standard is based on construction cost. If one looks at the trend of data collected in this study, the ratio of defect repair cost to construction cost decreases as the size of the construction cost increases. Considering these points, this study proposed an alternative plan of different deposit standards for each construction cost scale. The range of construction costs in the case studies was at a minimum of USD 7.74 million and a maximum of USD 825 million, and the alternative plans deduced within this range seem to be reasonable. However, this also means that the smaller the construction cost, the higher the repair cost ratio to construction cost. Therefore, in the case of low-cost construction houses that are not included in this study, one cannot rule out the possibility that a deposit of more than 3%, which is the current standard, may be required, as well as the revised standard proposed in

this study. Relatively small-scale builders often build houses with low-cost construction techniques, and these small builders not only have weak management conditions but also often have not been verified in terms of housing quality or ability to perform repairs. Therefore, in the future, it is necessary to collect data on low-cost construction houses to understand the level of repair costs and to adjust the deposit standards accordingly.

**Author Contributions:** Conceptualization, J.P. and D.S.; methodology, J.P. and D.S.; software, J.P.; validation, J.P. and D.S.; formal analysis, J.P.; investigation, J.P.; resources, J.P.; data curation, J.P.; writing—original draft preparation, J.P.; writing—review and editing, J.P. and D.S.; visualization, J.P.; supervision, D.S.; project administration, D.S.; funding acquisition, D.S. All authors have read and agreed to the published version of the manuscript.

**Funding:** This work was supported by the National Research Foundation of Korea (NRF) grant funded by the Korean government (2019R1A2C1009913).

**Data Availability Statement:** The data presented in this study are available on request from the corresponding author.

**Conflicts of Interest:** The authors declare no conflict of interest.

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
