# Peer review of "Defect Repair Deposit and Insurance Premium for a New Home Warranty in Korea"

_buildings, doi:10.3390/buildings13030815_

Round 1

Reviewer 1 Report

thank you for allowing me to review your work. It is very good in many ways, and clearly can be improved in others. Regarding the latter, it is possible to improve the discussion by emphasizing the reason for this over-dimensioning and also to incorporate, if there is any typology of what is the cause of the repairs, or the source of the failure, if it is in installations, finishes or general work. If this data exists, it would be very interesting to validate it for the conclusions and to review the difference with other countries.

Author Response

We authors have reviewed the paper thoroughly for improvements in the Discussion part as suggested by the reviewer. The contents of the Introduction, Review of Literature, and Discussion about the reason for excessive defect repair deposit were added.

Future research tasks were added: The authors also considered it very important to discuss the causes and problems of the housing defects suggested by the reviewer and to conduct a comparative study between countries. Since the construction, parts, structures, finishes, and facilities subject to housing defects are diverse, the authors also researched these subjects. So far, the analysis is limited to the cases of Korea, but in the future, as the reviewer suggested, the authors will conduct research about the differences in each country through exchanges with researchers in other countries.

Reviewer 2 Report

1. The paper deals with the problem of defects in newly built houses.

2. The topic is important. The paper covers the relevant social issues.

3. The paper has all necessary parts including introduction, main text, discussions and conclusions. The list of literature is impressive (66 items). 

4. The paper is more practical than theoretical. Although it seems an advantage, I suggest to add a broader theoretical context with relevant scientific issues both in the introduction and in the discussions.  

Author Response

As suggested by the reviewer, the content of the Introduction, Theoretical Considerations, and Discussion on the reason for the excessive defect repair deposit, which is the research topic, were added.

Reviewer 3 Report

The authors examine the important issue of addressing the costs incurred by defects in newly built houses in Korea. They reviewed the adequacy of the today’s 3% deposit calculation criteria through a case study of deposits and guarantee insurance premiums for housing defects. The results indicate that the deposit and guarantee insurance, according to the current standard, are excessive, at least for large housing projects. These levels of deposits and guarantee insurance premiums are compared with four proposed alternatives to improve the legislated practice.

This is an interesting applied research work, with results highlighted by example, comparative calculations and assessment, which however, need to be presented more carefully, with the necessary mathematical formulation.

The authors should address the following issues in an improved, revised version of their manuscript:

Abstract should become more concise and compact.

Lines 121-128: It is not clear from the presentation, if the insurance company may return a part of the 3% deposit to the payer (contractor or house owner?) after a number of years. Because insurance payments are not generally at a fixed premium. That is, a good contractor would be usually treated at better terms than a contractor with a bad history. (Lines 222-235 for the US).

Please explain further about similar practices adopted or expected to be adopted in Korea.

Lines 261-274 From the presentation it is not clear how the insurance premium is related to the 3% deposit in Korea: What is necessary here is to explain the relation by means of one or two mathematical equations, with the specific symbols clearly explained. For how many years needs the owner to pay the insurance premium etc.

Figure 1 is not easily readable. Fonts are too small.

Line 431, eq (1) How the authors are sure that the repair costs charged by the court was the maximum set by the deposit?   Please explain.

Figure 4: why the minimum ratios shown are higher than the maximum ratios? Please explain.

Conclusion section is OK, only minor improvement in its writing.

English need significant improvement throughout, preferably by a native speaker.

Author Response

Review 3  

Reviewer`s review

The authors examine the important issue of addressing the costs incurred by defects in newly built houses in Korea. They reviewed the adequacy of the today’s 3% deposit calculation criteria through a case study of deposits and guarantee insurance premiums for housing defects. The results indicate that the deposit and guarantee insurance, according to the current standard, are excessive, at least for large housing projects. These levels of deposits and guarantee insurance premiums are compared with four proposed alternatives to improve the legislated practice.

This is an interesting applied research work, with results highlighted by example, comparative calculations and assessment, which however, need to be presented more carefully, with the necessary mathematical formulation.

The authors should address the following issues in an improved, revised version of their manuscript:

1

Reviewer`s review

Abstract should become more concise and compact.

Author`s answer

As the reviewer pointed out, the Abstract of this paper was amended to be concise and clear (lines 8-20 of abstract).

2

Reviewer`s review

Lines 121-128: It is not clear from the presentation, if the insurance company may return a part of the 3% deposit to the payer (contractor or house owner?) after a number of years. Because insurance payments are not generally at a fixed premium. That is, a good contractor would be usually treated at better terms than a contractor with a bad history. (Lines 222-235 for the US).

Author`s answer

Below are the answers to the second question from the reviewer. 

a) As stated in lines 145 to 149 of Section 2.2 of this paper, generally, it is rare to put a housing defect repair deposit in cash in Korea. Instead, insurance premiums are paid while purchasing from a guaranty insurance company.

b) Therefore, since the policyholder does not pay a separate deposit, there is no refund even if the insurance contract is terminated.

c) Related contents are added in lines 149 to 150 of Section 2.2.

- Different policyholders’ insurance premium differences occur for the following reasons.

a) As the reviewer pointed out, in the United States and the United Kingdom, the insurance premium is determined through investigation and evaluation of the reputation of the housing company or housing quality.

b) On the other hand, in Korea, since insurance premiums are determined based only on the financial credit rating of the contractor, the housing construction company, there is no difference in insurance premiums according to insurance history.

c) Recently, as housing defects have become a social problem in Korea, it is known that guarantee insurance companies participate in housing quality inspections in cooperation with local governments and home construction companies.

d) However, insurance premiums are still not calculated considering the construction company’s reputation or the quality of the housing subject to guaranty.

e) As mentioned in lines 259 to 263 and lines 285 to 291 of Section 2.4, the authors agree to improve the above points, and the subject will be dealt with in the succeeding research.

3

Reviewer`s review

Please explain further about similar practices adopted or expected to be adopted in Korea.

Author`s answer

Please refer to the answers below for the third point by the reviewer.

a) Since the 2010s, the Korean government has been reorganizing and amending various laws and systems on housing defects.

b) The Korean government conducted research to establish criteria for housing defects in 2012 and 2013 and revised the related criteria in 2016.

c) The Korean government studied the warranty liability period system for housing defects in 2016 and reflected it in the Enforcement Decree of the Apartment Housing Management Act.

d) However, no research has been conducted at a government-level or institutional improvement regarding the housing defect repair deposit system.

e) The detail has been described in Section 2.4 (lines 259 to 263).

4

Reviewer`s review

Lines 261-274 From the presentation it is not clear how the insurance premium is related to the 3% deposit in Korea: What is necessary here is to explain the relation by means of one or two mathematical equations, with the specific symbols clearly explained. For how many years needs the owner to pay the insurance premium etc.

Author`s answer

Following is the answer to the fourth point.

a) The deposit for repairing housing defects in Korea is 3% of the construction cost, calculated as in equation (1). The guaranty insurance premium is calculated as in equation (2) by multiplying the deposit, the rate of the guaranty insurance premium, and the guaranty insurance period. Therefore, the guaranty insurance premium also changes when the deposit amount changes.

b) In addition, the insurance premium for housing defect repair in Korea is paid by the policyholder (mainly the home builder), who subscribes to the insurance upon completion of the house. Therefore, the homeowner does not separately pay the guaranty insurance premium.

c) The insurance period varies from 1 to 10 years for each specialized construction, such as reinforced concrete, finishing, facility, and landscaping, but insurance premiums are paid in a lump sum at the time of initial subscription.

d) This content was reflected in Section 2.4 (lines 276 to 284).

5

Reviewer`s review

Figure 1 is not easily readable. Fonts are too small.

Author`s answer

The fonts in Figure 1 are enlarged.

6

Reviewer`s review

Line 431, eq (1) How the authors are sure that the repair costs charged by the court was the maximum set by the deposit?   Please explain.

Author`s answer

Below is the answer to the sixth point.

- First, equation (1), stated in the paper submitted initially, explains that the defect repair deposit is set at 3% of the construction cost by the Korean Housing Act.

- The authors understood that the question by the reviewer was, “Is it reasonable to set the upper limit of the deposit only with the cost of repairing defects determined through the lawsuit?” Moreover, “If the deposit is claimed through the lawsuit filed again after the lawsuit or in another form, and if all of them are combined, will it exceed the specified deposit?” The answer would be as follows. 

   a) When writing a judgment, the court determines whether the homeowner’s repair claim exceeds the deposit. If the claimed defect repair cost exceeds the deposit, the defect repair cost is limited to the deposit amount and will be recorded in the precedent. However, among the precedents collected by the authors, there was no case where the repair cost exceeded the deposit.

   b) Generally, when the lawsuit is over, and the defect repair cost is paid to the homeowner, the warranty contract relationship is extinguished according to the guaranty insurance policy. In other words, the deposit, once paid, will not be paid again.

   c) In addition, since the completion of the lawsuit is approximately 5 to 10 years after the completion of the house, the legal statute of limitations has also expired. In other words, the homeowner’s legal claims are extinguished. Even if the homeowner files a lawsuit again, most of them are dismissed (in case the judgment is not made, and the case is decided to lose immediately because the legal requirements are insufficient, or judgment has proceeded but finally it is decided to lose because the legal requirements cannot be decided). As a result, the homeowner’s lawyer is reluctant to represent the case because the probability of losing the case is very high, and the lawyer’s reputation and performance suffer.

   d) Even if part of the warranty period remains since most defects have been compensated in the lawsuits already filed, duplicate claims cannot be made. Since the courts do not recognize such claims, the homeowners’ lawyers are aware of them, convincing their clients, the homeowners, that they cannot sue because there is no real benefit to them.

   e) Despite circumstances such as c) and d) above, there have been cases in the past where homeowners filed lawsuits directly without a lawyer, or lawsuits were filed on behalf of some homeowners. However, all of them lost, and as they were finally confirmed as precedents of the Korean Supreme Court, they had the effect of de facto law.

   f) As a result, it is difficult to file a lawsuit again once it has already been filed. Based on this fact, the authors judged that if the defect repair cost determined in one lawsuit did not exceed the deposit, the defect repair cost could be used as the criteria for determining the ceiling of the deposit.

   g) On the other hand, Korea’s housing defect repair guarantee insurance companies do not try to pay the deposit (insurance money) unless the lawsuit confirms it. Therefore, Korean homeowners cannot choose to file a lawsuit to recover the deposit. Therefore, numerous lawsuits have been filed in Korea, which has become a social issue.

7

Reviewer`s review

Figure 4: why the minimum ratios shown are higher than the maximum ratios? Please explain.

Author`s answer

Below is the answer to the seventh question.

a)   Figure 4 shows the guaranty insurance loss rate according to the guaranty insurance rate. As can be seen below, the minimum rate (red), average rate (green), and maximum rate (blue) at the bottom of Figure 4 all present the warranty insurance rates.

b) Equation (7) below calculates the guaranty insurance loss ratio. 

c) A guaranty insurance loss ratio refers to the ratio between premium income from the policyholders (mainly housing construction companies) and the paid insurance money (in case the developer and builder both become insolvent) to the insured (homeowners).

d) The insurance money corresponding to the numerator of the guaranty insurance loss rate calculation formula is the defect repair cost if the developer is insolvent, which is the amount determined by the lawsuit. As shown in Table 4, it is constant at $0.92 million.

e) Conversely, the guaranty payment corresponding to the denominator of the guaranty insurance loss ratio calculation formula changes according to the guaranty insurance rate.

f) The higher the rate, the higher the insurance premium, and the lower the rate, the lower the insurance premium. Therefore, the insurance premium is highest at the highest guaranty insurance rate and lowest at the lowest guaranty insurance rate.

g) For example, in the case of Plan A, as shown in Table 5, the insurance premium with the lowest rate is $1.88 million, and the insurance premium with the highest rate is $13.2 million.

h) When these values are substituted into equation (7), the insurance premium with the minimum rate applied is $1.88 million, and the insurance payment due to the builder’s insolvency is $0.92 million, so the guaranty insurance loss ratio is 48.94%, as shown below.

i) On the other hand, the insurance premium with the highest rate applied is $13.2 million, and the insurance payment due to the builder’s insolvency is $0.92 million, so the guaranty insurance loss rate is 6.97%, as shown below.

j) Likewise, since the numerator of the defect repair guaranty insurance loss ratio in equation (7) is constant and only the denominator changes, the loss rate tends to increase when the denominator, the insurance premium, decreases, and the loss rate tends to decrease when the insurance premium increases.

k) As shown below in Figure 4, the loss rate when the guaranty insurance premium rate is applied at the lowest rate is higher than when the highest rate is applied.

l) The explanation was added in Section 4.2 (lines 591 to 598).

8

Reviewer`s review

Conclusion section is OK, only minor improvement in its writing.

Author`s answer

The conclusion part has been supplemented as the reviewer suggested.

9

Reviewer`s review

English need significant improvement throughout, preferably by a native speaker.

Author`s answer

After amending the paper, the authors requested a correction from a native English speaker

Round 2

Reviewer 3 Report

You can publish the revised version

Author Response

Dear reviewer

I revised it according to your review, and the authors thank you for this.